# The record of the magnetic storm on 15 May 1921 in Stará Ďala (present day Hurbanovo) and its compliance with the global picture of this extreme event

Eduard Koči[1,2] and Fridrich Valach[1]

[1]Geomagnetic Observatory, Earth Science Institute, Slovak Academy of Sciences, Komárňanská 108, 947 01 Hurbanovo, Slovakia
[2]Slovak Central Observatory, Komárňanská 137, 947 01 Hurbanovo, Slovakia

**Correspondence:** F. Valach (geoffval@savba.sk)

**Abstract.** This paper deals with the most intense magnetic storm of the 20th century, which took place on 13–15 May 1921. Part of this storm was observed in magnetic declination and vertical intensity also at Stará Ďala, currently known as Hurbanovo. However, the sensitivity of the magnetometer was not determined there in the years when the storm occurred. Here, we estimated the sensitivity scale values on the base of data from before and after the studied event. The resulting digitized Stará

Ďala's data for 13–15 May 1921 are the main contribution of this work. The data were also put into the context of the records from other observatories. The overall picture of the geomagnetic field variations compiled from the observations by worldwide observatories, including Stará Ďala, suggests that the auroral oval got close to Stará Ďala and other European mid-latitude observatories in the morning hours on 15 May 1921.

## 1 Introduction

Due to its strong dependence on modern technologies, contemporary society has been increasingly posed with a threat to extreme space weather, of which geomagnetic storms represent an important aspect. In the present paper, we deal with an extraordinary magnetic storm, the cause of which was an extreme solar storm that was also known to the lay public at the time; it was commonly called the "Great Storm" or the "New York Railroad Storm" (Love et al., 2019; Hapgood, 2019). It manifested itself in the Earth's magnetic field on 13–15 May 1921. Part of its course was also recorded in former Czechoslovakia at

geomagnetic observatory Stará Ďala, formerly called Ógyalla, later Hurbanovo (Prigancová and Vörös, 2001). A few isolated observations of the magnetic declination were made at the Clementinum observatory in Prague (Swoboda, 1921) as well.

     The records of Stará Ďala for the study of this extraordinary geomagnetic phenomenon might be important, because the nearest observatory which at that time carried out at least limited geomagnetic observations (namely, the isolated observations of magnetic declination), Clementinum in Prague, was up to 369 km away from Stará Ďala. Other nearby observatories were

up to around 500 km away: in the south Pula (or Pola) on the Istrian Peninsula in Croatia – 473 km, in the west Munich in Germany – 493 km and in the north Swider in Poland – 520 km (see Hejda, 2007; Linthe, 2007). Furthermore, the relevant data from the Pula and Swider observatories are probably not even available. Towards the east, other geomagnetic observatories

were even more distant. Stará Ďala's observations were thus the only ones in a relatively large part of Europe. Unfortunately, the relevant hourly means in Munich are not available as recording the temporal variations of the geomagnetic field were discontinued during the years when the storm we are interested in occurred (Soffel, 2015). The closest observatory from which we used hourly means in this work is Seddin, Germany (Linthe, 2023b), 613 km away from Stará Ďala in a straight line. A few kilometres away, the Potsdam observatory (Linthe, 2023a) was also recording variations in the geomagnetic field at that time. The results of these two observatories were similar, but the analogue magnetograms from Seddin were more traceable and easier to read. Therefore, we preferred the data from Seddin in our study. The record from Clementinum (Swoboda, 1921) is not hourly means, but only a few instantaneous values.

Another example of a documented consequence of this magnetic storm was a fire at the telegraph station in Karlstad, Sweden, on 15 May 1921 at two o'clock in the morning of the local time (00:00 UT) (Em, 1921). The cause of the fire must have been geomagnetically induced currents caused by an extremely sharp change in the magnitude of the geomagnetic field, which during the violent variation could be up to ∼5000 nT/min (Kappenman, 2006). This value was impressive indeed when we consider that it is an order of magnitude more than on 13 March 1989, when the transformers of the electrical distribution network in Quebec were irretrievably destroyed (479 nT/min, according to (Kappenman, 2006)).

The most intense magnetic storms are often accompanied by auroras that are often observed at unusually low magnetic latitudes. A detailed study of historical aurora records is provided in (Silverman and Cliver, 2001), where the authors concluded that the lowest geomagnetic latitude at which an overhead (coronal) aurora was clearly observed during these geomagnetically disturbed days was about 40°. The exact value was 39.7°N, and it was an observation by Douglass (1921) from the Steward Observatory in Tucson, Arizona, on 15 May 1921 in the period between 03:30 UT and 05:30 UT. This value indicates the extreme equatorward extension of the auroral oval; that is, the auroral oval, at least for a short time and at least in a certain area of the Earth, got at the magnetic latitude of 40°, or maybe even under 40°.

The origin of the extreme geomagnetic activity that was observed on Earth has to be, of course, sought in solar activity. The study (Lundstedt et al., 2015) presented an attempt to understand the causative solar storm by means of a change in the complexity of the solar magnetic field. The active area with the Mount Wilson number 1842 was studied, which was first observed at the eastern limb of the solar disk on 8 May 1921. A method based on the Zeeman effect was used to obtain data on the solar magnetic field, and the presented model explained the topological changes of the field by approximating the solar magnetogram by a circular ring (torus), which underwent various changes: splitting, joining, twisting or undulation. The authors of the study expected that, by further developing their torus model, it would be possible to study the development of the active region number 1842 and perhaps arrive at an estimate of the energy that could be released during an extreme solar storm by means on the changes in the parameters of the complexity of the torus. Such further development of the model could be beneficial, but to our knowledge, it has not yet been implemented.

The first sudden commencement (SC), which was very likely related to the existence and development of sunspots in the 1842 region, caused a sudden increase in the horizontal component of the geomagnetic field by about 100 nT on 13 May 1921 at 13:06 UT – at low magnetic latitudes the increase was up to by 127 nT, the normalized global mean value was 89 nT. On the same day at 19:24 UT, another less pronounced SC event followed. On the following day, i.e. 14 May, a third SC occurred at

22:12 UT, for which the normalized value was 55 nT. According to International service on rapid magnetic variations (2023), which took the data on the examined SCs from (Mayaud, 1973), the duration of the main movements in these three storm sudden commencements were 1.5 min, 1.2 min, and 1.2 min, respectively; for some more details on the SCs evaluation see Mayaud (1973). We could assume that the key role in the subsequent extreme magnetic storm was played by two coronal mass ejections (CMEs), which manifested themselves as the first and third SCs amongst the mentioned SCs. It can be assumed that the first CME "cleared the way" through the interplanetary space for the second CME (Lundstedt et al., 2015), which could then hit the Earth in full force and cause the intense geomagnetic activity observed. Additionally, Hapgood (2019) assumes that also the CME related to the SC on 13 May at 19:24 UT contributed to clearing the way through the interplanetary space for the massive blow of the following CME.

This 13-15 May 1921 event is considered the most intense magnetic storm of the 20th century; in some parameters, it is even considered comparable to the Carrington storm of 2 September 1859 (Love et al., 2019). Comparing the May 1921 storm with the Carrington storm of September 1859 (Tsurutani et al., 2003), there is also the similarity that both of these superstorms occurred in a declining phase of the solar cycle, the solar cycles in both cases being mild and unremarkable (Lundstedt et al., 2015).

It should be noted here that the Carrington storm is usually considered a prototype of the most extreme solar storm that could threaten the Earth. However, this assumption may not be correct at all, since flares have been observed on slowly rotating Sun-like stars that released 1000 times more energy than a Carrington flare (Maehara et al., 2012).

To contribute to a better understanding of extreme geomagnetic events, we processed records of magnetic declination and vertical intensity for the magnetic storm of May 1921 that we discovered in the archives of the Stará Ďala observatory. The presentation of these processed, although not quite complete magnetograms are the main goal of our work.

In addition, we compare these unique data with data from other observatories. These additional data are: (1) little-known observations of magnetic declination from the Clementinum observatory in Prague (Swoboda, 1921) supplemented by information about auroras from various observation sites in former Czechoslovakia, which we will reproduce in this article, and (2) data from another European observatory, Seddin (in Germany).

It is likely that the disturbances recorded at individual mid-latitude observatories were not caused only by the strengthened ring current. An important co-responsibility for the violent geomagnetic variations might be also assigned to the expansion of the auroral oval. In connection with this storm, this idea was already pointed out by Love et al. (2019). A secondary goal of our work is to use data from various observatories in different parts of the globe, including data from Stará Ďala, to show the approximate extent of the auroral oval during the New York Railroad Storm. At the same time, we intend to demonstrate that the data from Stará Ďala fit into the overall picture of the magnetic storm of May 1921.

## 2 The magnetograms of the storm of May 1921 from the Stará Ďala observatory

In the following part of the article, we present the record of the storm of 13–15 May 1921 according to magnetograms for magnetic declination and vertical intensity that were found in the archive of the Stará Ďala observatory (for the coordinates

**Table 1.** The coordinates of the Stará Ďala geomagnetic observatory together with nearby observatories Clementinum (Prague) and Seddin (in Germany) for epoch 1921.4.

|  | Geographic | | Quasi-dipole (QDP) | |
| --- | --- | --- | --- | --- |
|  | Latitude | Longitude | Latitude | Longitude |
| Stará Ďala | 47.87°N | 18.19°E | 42.98°N | 94.09°E |
| Clementinum | 50.08°N | 14.42°E | 45.85°N | 91.79°E |
| Seddin | 52.28°N | 13.01°E | 48.43°N | 91.45°E |

of the observatory, see Table 1). We will respectively deal with (1) the description of the magnetograms (Sect. 2.1), (2) the determination of the scale for declination and vertical intensity (a description in detail of the calibration procedure, Sect. 2.2), and (3) the resulting calibrated and digitized data drawn from the magnetograms (Sect. 2.3).

## 2.1 The record found in the archive of the observatory

The Stará Ďala geomagnetic observatory started operating under the previous name Ógyalla (in another spelling Ó Gyalla) in 1894, when the magnetic measuring instruments from the Buda observatory were transferred to this new observation station. The Buda observatory's observations gradually began to be affected and spoiled by artificial disturbances in the geomagnetic field due to the expansion of the city; Buda is a part of the Hungarian metropolis Budapest.

During the period for which we processed the records in this study, the geomagnetic field was recorded using the Carpentier magnetograph with photographic recording. The necessary absolute measurements of the horizontal intensity and magnetic declination were performed with a Wild magnetic theodolite (Ochabová and Ochaba, 1977; Wienert, 1970, p. 54-81). The magnetic inclination was determined by means of an earth inductor (Wienert, 1970, p. 95-101), which was built in 1905 by A. Büky using parts of Wild and Mayerstein theodolites. In 1911, observers decided to end absolute measurements of inclination with this instrument, because it was decided that they no longer had much value, as the vertical variometer had become unreliable (Ochabová and Ochaba, 1977).

However, as we will show below, the recording of vertical intensity variations continued, but without determining the baseline values and carrying out calibrations. We must take the information about the unreliability of the variometer into account; we do not have more detailed information about the nature of that unreliability, but we assume that even if the data obtained from the magnetogram for the vertical intensity were not quite accurate, they are perfectly sufficient for the qualitative interpretation of the extreme event of May 1921.

Other imperfections of the records are: missing information on the horizontal intensity; no description of time marks (but time marks themselves – regularly recurring short interruptions of the baseline – are visible); missing absolute measurements at that time, even for magnetic declination; and going out of measurement range in both recorded geomagnetic elements at the time of the most intense disturbed geomagnetic field.

In Supplement (Fig. S1), the reader can find the scans of the records of the storm during the period 13 to 15 May 1921 that we discovered in the archive of the Stará Ďala observatory. Unfortunately, we could not reliably identify the photographic record for the second part of 15 May; therefore, our data from Stará Ďala ends shortly after 10:00 UT.

## 2.2 Scale for the recorded variations on the magnetograms

To make the photographic record of the geomagnetic field valuable for studying, we first needed to describe the time axes of
the magnetograms made during the extreme storm of May 1921. In addition, we had to determine the instrument's scale factors with which the magnetic declination and the vertical component were recorded on photographic paper.

   Time instants were assigned to the time marks by comparing Stará Ďala's records with the records made by some other observatories, in particular by Tuscon and Honolulu, whose observations are reproduced by Kappenman (2006) and described by Love et al. (2019) and Hapgood (2019). The instants of global phenomena in the geomagnetic field were compared, such as
sudden commencements (SCs) and the variations caused by the ring current. We also know that the then device worked in the so-called one-centimetre recording mode, which means that one centimetre on the timeline corresponds to one hour. In fact, the recording in the variation device was made on photographic paper, which was moved at a speed of 1 cm per hour using a clockwork machine.

   A more difficult task was to determine the sensitivity of the device, i.e. how much change in declination (in minutes of arc)
and Z component (in nanoteslas) corresponds to a change in length unit (millimetre) in the direction of the vertical axis. This section is devoted to the solution to this task.

### 2.2.1 The method used to determine the declination scale

Absolute geomagnetic measurements to determine the sensitivity of the recording equipment were not carried out at the Stará Ďala observatory in the years when the studied extreme storm occurred. Therefore, we had to settle for an estimated scale value
in our study. For this estimation, data from the surrounding periods (before and after the studied event) were used, in which it was possible to determine the scale. In doing so, we assumed that the scale did not change significantly between neighbouring periods with the known scale. As we will show, this condition was met sufficiently well.

   A combination of the two following ways was used to determine the scale:

A In the first technique, we calculated the scales from the data for the period before the studied event and after the event.
A part of the declination data was found in the observatory's yearbooks; in some other cases, the declination values were inscribed on the magnetograms by observers at the time. We took the scale as the ratio of the variation data of the declination in arc minutes and the corresponding change on the graphic record (magnetogram) in millimetres.

B As a second way, we compared the hypothetical quiet diurnal variation in 1921 near the summer solstice (in arc minutes) with the variation recorded on the magnetograms (in millimetres). That diurnal variation might represent the Sq variation
for perfectly quietened geomagnetic activity. The hypothetical solstice variation of the declination was estimated based on the periods near the summer solstices in the surrounding years. We took the summer periods because the Sq varia-

tion is most pronounced in those seasons, which should be advantageous for reducing the uncertainty in the estimated declination scale.

### 2.2.2 The resulting scale for the declination in May 1921

Following procedure A, we calculated the scale factor for some periods before 1921 and after 1921. For this purpose, magnetograms were selected (Table 2) that satisfy two requirements: (1) they distinctly displayed the course of declination and enabled reading the extent of variations in millimetres, and (2) there was information available about the extent of the corresponding variations in arc minutes. For the years 1913–1919, we obtained the scale factor $1.248'/\text{mm} \pm 0.038'/\text{mm}$, and for years 1927 and 1928, we calculated the value $1.176'/\text{mm} \pm 0.017'/\text{mm}$. In this paper, the central value is represented by a

median and its uncertainty is quantified by the median of absolute deviations (MAD). We adopted this kind of statistic because it is less affected by outliers and asymmetrical data compared to commonly used mean and standard deviation. For the sake of completeness, we note that for the data until 1918, the observatory name was Ógyalla, instead of Stará Ďala (nowadays Hurbanovo). In 1929, the scale was substantially changed (the found values were $0.632'/\text{mm}$ and $0.693'/\text{mm}$); the data since 1929 were thus not included in this study. The scale did not significantly change between 1913–1919 on one side and 1927–1928 on

the other side. The scale factor calculated on the base of all items in Table 2 is $1.199'/\text{mm} \pm 0.013'/\text{mm}$.

The assumption about unchanged scale between 1913 and 1928 could also be verified by comparing diurnal variations during magnetically quiet days. As mentioned, the periods near summer solstices are the best for this purpose. Table 3 lists the differences between daily maxima and minima for such quiet days. Before 1921, in 1921 and after 1921, the differences were $8.80\,\text{mm} \pm 0.25\,\text{mm}$, $8.64\,\text{mm} \pm 0.05\,\text{mm}$, and $8.45\,\text{mm} \pm 0.08\,\text{mm}$, respectively. The device had thus same parameters in

1921 as before and after that year. Considering all the data in Table 3, the Sq variation in declination near the summer solstice was recorded as $8.55\,\text{mm} \pm 0.05\,\text{mm}$.

Before and after 1921 (Table 4), the quiet diurnal variation (daily maxima minus minima) for selected days was $10.3' \pm 0.1'$ and $10.0' \pm 0.0'$, respectively; in the latter result, the estimated error of the centre value was less than the accuracy with which the result is expressed. As expected, these two values differ just a little. Considering all data in Table 4, the Sq variation gives

$10.0' \pm 0.0'$.

To calculate the scale on the base of procedure B, we employed the data on the Sq variations in arc minutes and millimetres listed in Tables 3 and 4. Taking the median values in these tables, one obtains the scaling factor equal to $1.17'/\text{mm} \pm 0.01'/\text{mm}$.

The final result for the declination scale, which is $1.18'/\text{mm} \pm 0.01'/\text{mm}$, was obtained as a weighted average of the scale factors yielded by procedures A and B. In the averaging process, the inverted squared values of the uncertainties were employed

as weights.

### 2.2.3 The method used to determine the vertical intensity scale

In the case of vertical intensity, too, we were compelled to rely on an estimated value of the scale factor. The nearest time interval to 1921 for which absolute values of the vertical intensity are available is the years 1909 and 1910. In 1911, the absolute device for the vertical intensity became unreliable, and the absolute measurements ceased. The Stará Ďala observatory waited

**Table 2.** Scale factors for magnetic declination determined in selected days.

| Date | Scale ($'$/mm) |
|------|------|
| Prior to 1921 | |
| 22.05.1913 | 1.248 |
| 06.06.1913 | 1.147 |
| 07.06.1913 | 1.136 |
| 02.06.1918 | 1.207 |
| 23.06.1918 | 1.308 |
| 05.10.1919 | 1.349 |
| 17.10.1919 | 1.349 |
| 22.10.1919 | 1.376 |
| After 1921 | |
| 07.01.1927 | 1.000 |
| 19.06.1927 | 1.149 |
| 20.06.1927 | 1.111 |
| 29.06.1927 | 1.220 |
| 10.06.1928 | 1.176 |
| 14.07.1928 | 1.221 |
| 16.07.1928 | 1.190 |

three decades (until 1941–1942) for the restoration of the measurements of the vertical intensity. Because the 1940s are too distant from 1921, we were limited to utilising the data of the years 1909 and 1910. Moreover, we revealed that the scale for the vertical intensity changed significantly at the end of June 1909; therefore, only the data from the second half of 1909 might be employed.

Considering the more than ten-year distance, we suspected that the scale determined for the second part of 1909 and 1910 did not keep its validity as far as until 1921. Thus, we needed to modify the approach we used for the declination. First, we determined the scale factor for the period from July 1909 to December 1910: For large variations during geomagnetically disturbed days, we found the changes in the vertical intensity expressed in nanoteslas (data provided in the observatory yearbook). These data were then compared with the corresponding variations in millimetres caught on the magnetograms.

In the next step, we estimated a ratio between the scale valid before 31 December 1910 and that in the year 1921: We compared variations during days near the summer solstices (the variations expressed in millimetres) on thoroughly selected magnetograms in both periods. Those records were selected in which the light trace on the magnetogram (photo paper) was clear and thin, to allow obtaining values with as little uncertainty as possible. Because in 1909 and 1910, the record-keeping medium (photographic paper) was regularly replaced at 12:30 UT, we decided to compare the increase in the vertical intensity between 13:00 UT and 17:00 UT. The reason for this decision is that relatively significant growth in vertical intensity is typical

**Table 3.** Daily maxima minus minima (in millimetres) on analogue photo-paper records of magnetic declination during selected quiet days.

| Date | Variation (mm) |
| --- | --- |
| Prior to 1921 | |
| 22.05.1913 | 8.25 |
| 06.06.1913 | 9.50 |
| 07.06.1913 | 8.80 |
| 05.07.1917 | 9.20 |
| 19.06.1919 | 7.80 |
| In 1921 | |
| 01.06.1921 | 8.72 |
| 18.06.1921 | 8.35 |
| 21.06.1921 | 8.30 |
| 22.06.1921 | 8.78 |
| 24.06.1921 | 8.64 |
| 10.07.1921 | 8.51 |
| 18.07.1921 | 8.74 |
| After 1921 | |
| 19.06.1927 | 8.70 |
| 20.06.1927 | 8.10 |
| 29.06.1927 | 8.20 |
| 10.06.1928 | 8.50 |
| 14.07.1928 | 8.60 |
| 16.07.1928 | 8.40 |

for this time interval at Stará Ďala during the quiet days. In the end, the scale for 1921 will be estimated as the ratio between the variations before 31 December 1910 and those in 1921 multiplied by the scale determined on the basis of the disturbed days in 1909 (only the second half of the year) and 1910.

The scale factor before 31 December 1910 might hypothetically be also obtained based on diurnal variations instead of the geomagnetic disturbances. Such an approach, however, led to large statistical errors in the case of the vertical intensity, and

therefore we omitted it here.

### 2.2.4   The resulting scale for the vertical intensity in May 1921

Based on geomagnetically disturbed days (Table 5), we determined the scale factor $3.11\,\mathrm{nT/mm} \pm 0.02\,\mathrm{nT/mm}$. This value is valid for the second half of 1909 and the whole year 1910.

**Table 4.** Daily maxima minus minima of magnetic declination during selected quiet days before 1921 and after 1921, respectively.

| Date | Variation (′) |
|---|---|
| Prior to 1921 | |
| 13.07.1911 | 10.6 |
| 22.05.1913 | 10.3 |
| 06.06.1913 | 10.9 |
| 07.06.1913 | 10.0 |
| 18.06.1913 | 10.0 |
| After 1921 | |
| 19.06.1927 | 10.0 |
| 20.06.1927 | 9.0 |
| 29.06.1927 | 10.0 |
| 10.06.1928 | 10.0 |
| 14.07.1928 | 10.5 |
| 16.07.1928 | 10.0 |

**Table 5.** Scale factors for the vertical intensity determined for selected days.

| Date | Scale (nT/mm) |
|---|---|
| 03.07.1909 | 3.02 |
| 21.07.1909 | 3.15 |
| 21.09.1909 | 3.11 |
| 30.09.1909 | 3.02 |
| 27.03.1910 | 3.06 |
| 31.03.1910 | 3.55 |
| 27.04.1910 | 3.12 |
| 02.05.1910 | 3.25 |
| 02.05.1910 | 3.47 |
| 22.08.1910 | 3.06 |
| 27.10.1910 | 3.07 |

During geomagnetically quiet days near the summer solstices in 1909 and 1910, we investigated an increase in vertical intensity between 13:00 UT and 17:00 UT. Its size on the photographic papers was 2.90 mm $\pm$ 0.05 mm (see the upper part of Table 6 for the individual values). Similarly, near the solstice in 1921, the growth of the vertical intensity between 13:00 UT and 17:00 UT was 5.55 mm $\pm$ 0.07 mm (bottom part of Table 6). Applying the simple procedure outlined in Sect. 2.2.3, we obtained the scale factor value for the vertical intensity in 1921, equal to 1.63 nT/mm $\pm$ 0.04 nT/mm.

The diurnal variations are caused by the Sq currents, and the amplitudes of those variations might depend on the solar activity conditions. For instance, Owolabi et al. (2022) showed a strong dependence of Sq intensity on F10.7. Because no data on F10.7 are available for the studied period, we used the sunspot number to describe solar activity in the years 1909-1910 and 1921. The two periods fell into the declining phase of the two consequent solar activity cycles. The sunspot number did not differ substantially between the selected days in 1909-1910 and 1921. Namely, the sunspot number was between 10 and 45 during the days in 1909 and 1910, with one exception on 25 July 1909, when the sunspot number was 153; and in 1921, the values were from 18 to 42 (Royal Observatory of Belgium, Brussels, 2023). In our data set (Table 6), we found only a very weak or even no relationship between the vertical intensity variations and the sunspot number. Thus, we assume that our determination of the scale factor was not influenced substantially by different conditions in solar activity in those two periods separated by approximately 11 years.

### 2.3 Stará Ďala's records of the May 1921 storm – the resulting time series

After the scale was determined for both the magnetic declination and the vertical intensity in the original photo-paper records, we detemined the five-minute data (i.e., five-minute means) from the magnetograms. For this purpose, we imitated the method commonly used in determination the hourly averages by means of a glass scale. The mean value was determined using an imaginary horizontal straight line, which was put on the magnetogram so that we were making equal the areas between the trace being scaled and the horizontal line (McComb, 1952, pp. 177–178). Where the magnetograms were incomplete, five-minute means could not be determined; instead, we read momentary values, provided at least some short part of the traces on the records were distinct. In some potentially interesting local extrema, we supplemented the five-minute means with momentary values even if the five-minute data were available. The resulting data series are graphically displayed in Fig. 1, and the corresponding data files are stored in Supplement. Figure 1 also indicates three SCs found for the studied event by Mayaud (1973). The first SC, which Mayaud classified as "very distinct s. s. c.", is visible in the declination as a sharp variation. The remaining two SCs were less dictinct.

### 3 The Stará Ďala record in the context of reports from other observatories

Comparing the Stará Ďala record with the records of the geomagnetic field made by other observatories might make Stará Ďala's magnetogram a piece in the complicated mosaic of that spectacular storm of May 1921. In the following sections, we first compare Stará Ďala's observations with the data of the nearest observatories that were in operation at that time.

**Table 6.** Increase in the vertical intensity in millimetres between 13:00 UT and 17:00 UT. The increase was read on magnetograms on selected geomagnetically quiet days in July 1909 and near the summer solstices of 1910 and 1921.

| Date | Increase (mm) |
|---|---|
| In 1909–1910 | |
| 16.07.1909 | 3.05 |
| 25.07.1909 | 2.90 |
| 23.05.1910 | 2.90 |
| 04.06.1910 | 3.45 |
| 06.06.1910 | 2.20 |
| 07.06.1910 | 2.70 |
| 14.06.1910 | 2.85 |
| 30.06.1910 | 3.20 |
| 01.07.1910 | 2.80 |
| In 1921 | |
| 23.05.1921 | 5.70 |
| 25.05.1921 | 5.00 |
| 01.06.1921 | 5.50 |
| 05.06.1921 | 5.55 |
| 18.07.1921 | 5.85 |

Subsequently, we will examine some consistent features in the course of the geomagnetic field with observatories at various locations over the globe; such a comparison could indicate, for instance, the likely extent of the auroral oval during the storm.

### 3.1  Comparison with the observations in Prague and Seddin

The nearest place to Stará Ďala from which some geomagnetic records (Swoboda, 1921) are available during May 1921 is the Clementinum observatory in Prague (Table 1). The recorded quantity was magnetic declination, which was then only observed

by eye, usually three times a day. However, during geomagnetic storms, the frequency of the observations increased; this way the geomagnetic disturbance shown in Fig. 2 was caught.

The essential features of the declination in the Prague record (Swoboda, 1921) on 15 May 1921 can be summarized as follows: until 00:00 UT on May 15, the geomagnetic field was quiet (declination between ca $-6°20'$ and ca $-6°40'$); at 06:00 UT, an increase appears ($-5°59'$); at 07:00 UT the first maximum occurred ($-5°29'$); the value measured at 13:00 UT was

then similar to those in the quiet period ($-6°27'$); at 18:30 UT the second maximum was recorded (more pronounced than the first one, the value being $-5°13'$); at 19:00 UT, the extreme value persisted ($-5°21'$); and at 20:00 UT, the geomagnetic field already quiet ($-6°24'$).

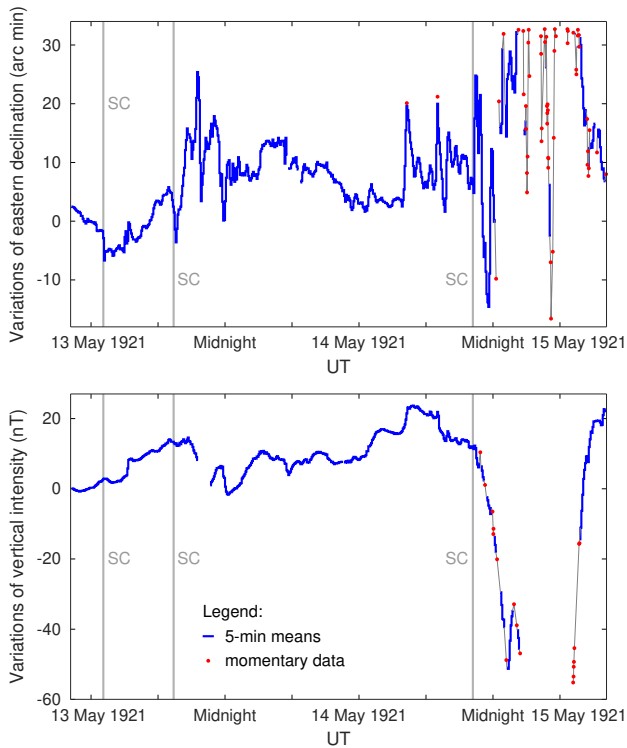

**Figure 1.** The records of magnetic declination and vertical intensity in Stará Ďala for the storm of 15 May 1921. The five-minute means, as well as some important momentary values are displayed. The occurrences of SCs on 13 May at 13:06 UT and 19:24 UT and on 14 May at 22:12 UT are also indicated (according to Mayaud, 1973).

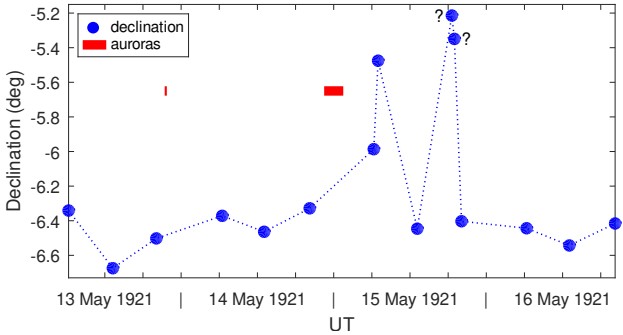

**Figure 2.** Geomagnetic disturbance recorded in magnetic declination on 13–16 May 1921 at observatory Clementinum in Prague. The times are also indicated when auroras were observed in former Czechoslovakia. The lines connecting the observations is added only to guide the reader's eyes. The question marks indicate the values of whose correctness we are not entirely convinced. (After the data by Swoboda, 1921).

In addition, Swoboda (1921) informs that northern lights were observed from several places in former Czechoslovakia. Based on information from the state meteorological institute, he writes: 'In the nights from 13 to 14 May and from 14 to 15 May 1921, beautiful polar lights were observed at several places of the republic. The aurora was well visible because the weather was clear, the air was particularly transparent, and on the second night, the Moon set early. On May 13, the sky looked like it was lit by large reflectors, the light was whitish or even white, and it changed to grey-green up to deep green colour. Later, following three times one after another, blood-red beams lightened, each holding for about one minute. (Observation places: Brno, Dobrovice, Jaroměřice, Rakovník, Roudnice, Rokytnice, Rosice u Chrudimi [quasi-dipole (QDP) latitudes of these places are between 44°37′N and 46°14′N]. Observed from 22 h 25 min until 3/4 to 23 [21:25 UT – 21:45 UT]). The following night (from 14 to 15 May), when the Moonlight did not disturb, the phenomenon exhibited blood-red radiation, which was generally reckoned as the blare from some distant fire. (Observation places: Litoměřice, Lnáře u Blatné, Velké Meziříčí, Milovice, Plzeň [QDP latitudes of these places are between 44°53′N and 46°24′N] and others, the beginning of the observations between 23 h 30 min and 1 h 20 min, end at about 1/2 past 2 [i.e., beginning between 22:30 UT and 00:20 UT, end at ca 01:30 UT])'.

The observation times for auroras seen from Czechoslovakia are in good agreement with the observation reported by Hapgood (2019) in his article citing Wilkens and Emde (1921). They wrote that the aurora was observed in Bremen (northern Germany) between 21:10 UT and 22:20 UT on 13 May. Also, an aurora was observed on 15 May in Breslaw (today Wroclaw, southwestern Poland) between 00:20 UT and 01:30 UT. Thus, the aurora observations from Czechoslovakia are a credible contribution to the list of aurora observations published in (Hapgood, 2019); these observations cover a part of Europe that Hapgood's list does not contain.

At that time, the observatory in Prague was not recording the geomagnetic elements other than declination. Therefore, we also used data from another, a little more distant European observatory Seddin for comparison with the Stará Ďala magnetograms. For its coordinates see Table 1. The analogue magnetograms displaying the geomagnetic storm of May 1921 recorded by the Seddin observatory (north, east, and west components of the geomagnetic field) are shown in Fig. 3. To better outline the gross features of the event, we also provide Fig. 4, which shows the hourly averages of horizontal intensity, declination and vertical intensity.

The first maximum recorded in Prague at ca 07:00 UT agrees with what can be seen in the analogue magnetogram recorded in Seddin (see the central part in Fig. 3; the variations in the east component approximately correspond to the variations in the declination). It also agrees with the large increase in declination seen in Fig. 4 on 15 May in the morning hours. Moreover, that first Prague's maximum indicates that within the period when the declination in Stará Ďala went out of the scale of the recording apparatus, the variation in Stará Ďala was eastern. The second maximum that was reported in Prague at ca 18:30 UT cannot be seen in the magnetogram of the relatively nearby Seddin. It seems that this maximum was just a local phenomenon. We also admit the two unexpectedly extreme data forming the second maximum might be erroneous; the cause of the error, however, has remained unrevealed for us. We did not manage to compare Prague's second maximum with data of Stará Ďala because its photo-paper magnetogram from the second part of 15 May could not be reliably identified.

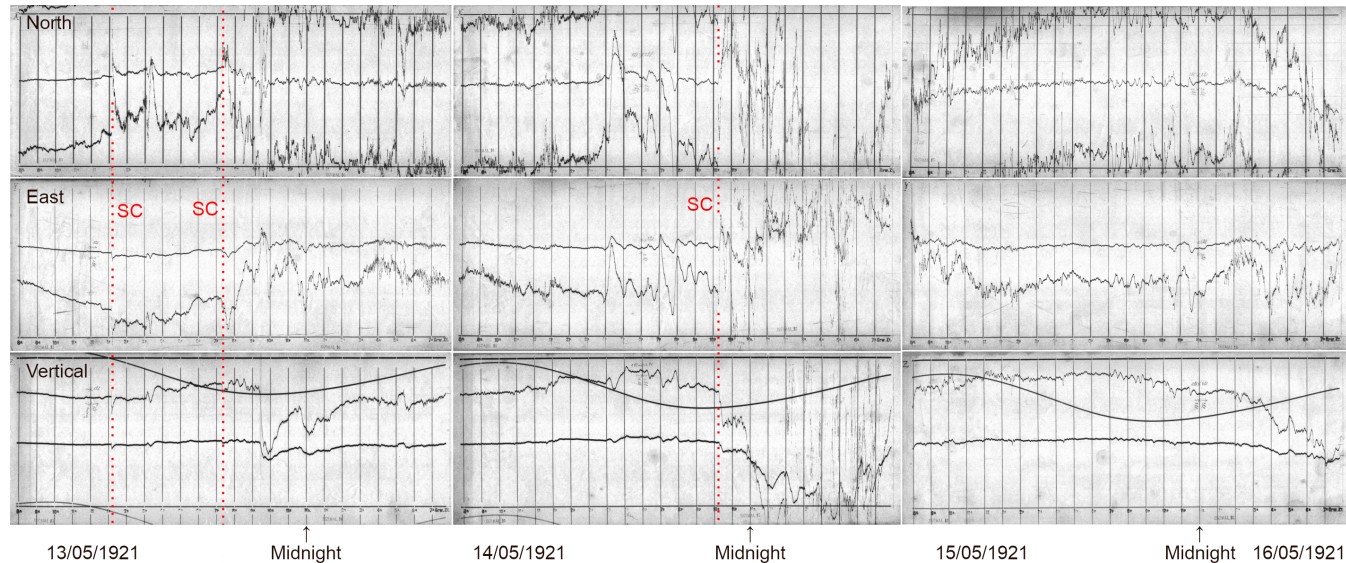

**Figure 3.** Analogue magnetograms displaying the geomagnetic storm of May 1921 as recorded at the Seddin observatory. The following elements are displayed: north component (X, upper panel), east component (Y, middle panel), and vertical intensity (Z, bottom panel). The sudden commencements listed in (Mayaud, 1973) are indicated on 13 May at 13:06 UT and 19:24 UT, and on 14 May at 22:12 UT. In addition to the displayed SCs, Mayaud (1973) reports a smaller SC on 16 May at 01:24 UT, but we do not show this SC in the picture nor discuss it in this paper as it is beyond the period we are focusing on. (The scans of the magnetograms were obtained from the archives of the Niemegk observatory.)

Besides the magnetic declination and vertical intensity, which were both recorded in Stará Ďala too, Fig. 4 for Seddin also displays the horizontal component of the geomagnetic field. Such a piece of more complete information gives an idea about the extent of the geomagnetic storm. The violent drop of the hourly means of the horizontal intensity by ca 850 nT, which occurred
in the early morning hours, is indeed an extreme variation for mid-latitudes.

Figs. 3 and 4 also show the occurrences of three extremely large sudden commencements (SCs), the first two of them occurred on 13 May at 13:06 UT and 19:24 UT, and the third occurred on 14 May at 22:12 UT. The first two SCs may be related to the arrival of two coronal mass ejections (CMEs). By passing through interplanetary space, these two CMEs likely reduced the interplanetary plasma density and allowed a third CME to move quickly towards Earth. That third CME
caused the SC on 14 May and originated subsequent extreme geomagnetic activity (Hapgood, 2019). The SCs were large, and the corresponding variations were distinctly recognisable in the north component as well as in the other geomagnetic field components.

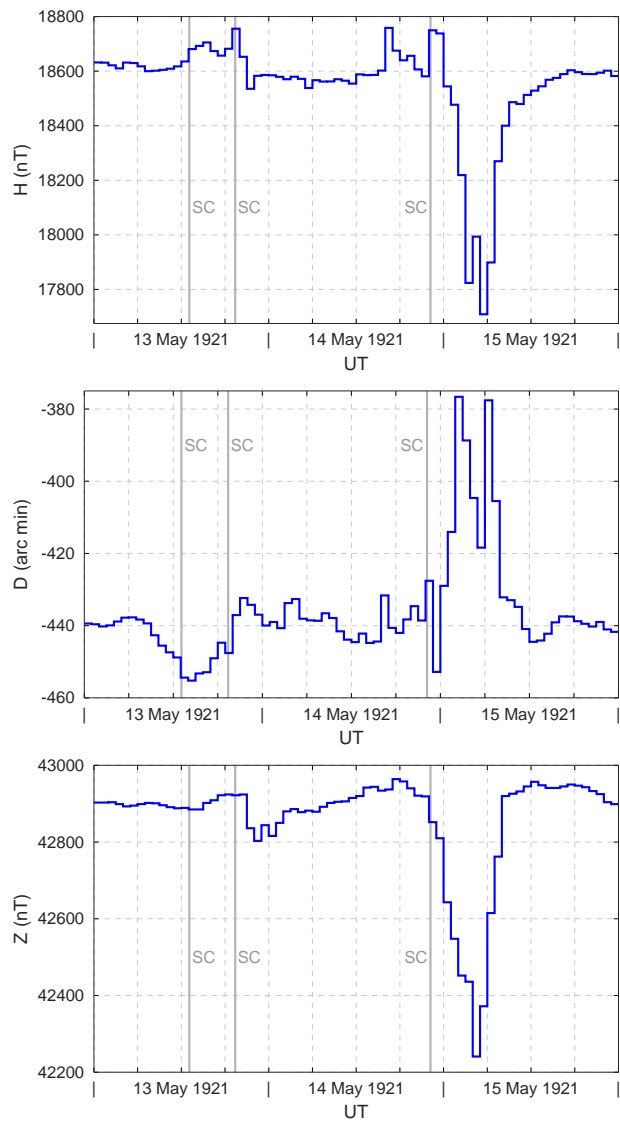

**Figure 4.** Hourly means of the geomagnetic elements from the observatory Seddin for the period 13-15 May 1921. The following elements are displayed: horizontal intensity (H, upper panel), declination (D, middle panel), and vertical intensity (Z, bottom panel). The times of the occurrences of the sudden commencements according to Mayaud (1973) are also indicated.

## 3.2 Comparison with observations at different places on the globe

In this section, we display the geomagnetic field variations at different places on the globe using a polar coordinate system. As variations, we took the differences between the hourly values of the individual geomagnetic elements and their respective quiet values, the latter being equal to the values during the quiet period before the geomagnetic storm started. Such a representation of the data can help to create an image of the manifestation of the storm in different parts of the world, which may give a cue for studying some electric currents responsible for the geomagnetic variations.

In this part of the paper, our study is limited to the most pronounced variations in horizontal intensity, declination and vertical intensity. This limitation is applied here due to that large variations are typical for the closeness of the auroral oval. Moreover, we were only interested in those large variations that were observed in mid-latitudes. The thing is that we want to describe situations where the auroral oval extended far equatorwards from its typical position.

### 3.2.1 Data from some worldwide observatories and the methodology used

The time evolution of the horizontal intensity, magnetic declination, and vertical intensity in the period 13–15 May 1921 from observatories from different parts of the globe we studied on the base of a series of graphs with the one-hour time step. The data utilised here were all the hourly means of the geomagnetic field available for the period in question via the webpage of the World Data Center for Geomagnetism, Kyoto (2023). The hourly means of the Seddin observatory were obtained from the archive of the Niemegk observatory.

The hourly means for Stará Ďala were calculated from the five-minute means. We did so even for incomplete data sets, providing there was at least a reliable five-minute mean available. This, on one side, limits the accuracy of the hourly averages; on the other side, however, such an approach lessens the data gaps just in the most interesting sections, where the geomagnetic field is disturbed the most. In case of doubts in the interpretation of such data, it is always possible to consult the original five-minute data in Fig. 1 or in Supplement.

The graphs are drawn in a polar coordinate system. The origin of this system is the north geomagnetic pole based on the QDP (see, e.g., Laundal and Richmond, 2016) coordinates valid for the middle of 1921, the radial coordinate is the QDP latitude, and the angular coordinate (azimuth) is the magnetic local time. The reader can find a complete time series of the polar graphs displaying the variations of the horizontal intensity, declination and vertical intensity in Fig. S2. In Sect. 3.2.2, graphs for the horizontal intensity and declination during selected crucial periods will be shown, too.

The series of images in Supplement (Fig. S2, as well as those few selected images in Sect. 3.2.2) are made to show the situation relevant to the northern hemisphere. A positive variation in declination is displayed with an arrow pointing east, and an increase in horizontal intensity is shown with an arrow pointing toward the pole. Variations from a few observatories from the southern hemisphere were also integrated into the pictures; these variations are drawn differently, and the orientation of the arrows is adjusted as the respective variations would appear in the northern hemisphere if they were the result of currents in auroral ovals belonging to the northern hemisphere. It means that, for the data in the southern hemisphere, we displayed a positive variation in declination with an arrow pointing west (i.e. in the opposite direction than in the northern hemisphere),

and an increase in horizontal intensity is shown with an arrow pointing toward the pole (i.e. as in the northern hemisphere). Horizontal intensity variations are shown by arrows (red for Stará Ďala, blue for other Northern Hemisphere observatories, and empty for southern hemisphere observatories) in the radial direction; the direction towards the pole indicates growth, and the lengths of the arrows indicate the magnitude of the variations. Variations in declination are shown by arrows oriented tangentially to the circles of latitude (same colour code as mentioned above), with a counter-clockwise arrow indicating eastern variation. Variations in vertical intensity are shown by arrows pointing up or down (a down arrow represents an increase in the vertical intensity), with the magnitude of the variation indicated by the size of the symbol (data from the northern hemisphere is represented by black symbols and from the southern hemisphere by grey symbols). The parallel drawn in green represents the QDP latitude at which the most equatorward observed overhead aurora was reported during the May 1921 storm.

### 3.2.2 The position of the equatorward boundary of the auroral oval in May 1921 and some related currents

This section is devoted to a brief interpretation of the time evolution of the geomagnetic field variations caught in the graphs described in the previous text (Sect. 3.2.1). The observations made at Stará Ďala are a part of them.

The decrease in horizontal intensity, as long as it occurred globally at all observatories, can be interpreted as a geomagnetic variation caused by the ring current. On the other side, changes in the horizontal intensity that occurred locally, not around the whole globe, were caused by electric current systems other than the ring current. We assume here that the equatorward edge of the auroral oval reached as far as mid-latitudes during this extreme storm. Therefore, we believe that those local variations were due to the current system of the auroral oval.

In the studied part of the Earth with a magnetic inclination of ~70°, we excluded field-aligned currents (FACs) from our considerations as a direct cause of the observed sharp variations of the geomagnetic field. In the spot below the place in the auroral region where a FAC enters or leaves the ionosphere, nor in the close vicinity of that spot, the ground-based observatories cannot observe the magnetic field generated by the FAC. Close to the equatorward boundary of the auroral oval, we must therefore look for some other current responsible for the observed deflection of declination. It might be a certain amount of Hall current in the north-south direction; such north-south Hall currents appear between pairs of FACs separated in the east-west direction.

We studied the time evolution of the geomagnetic variations on the base of hourly means. Some short-lasting variations thus might be smoothed or completely lost. Nonetheless, the most pronounced variations, which are the main focus of this study, must be visible also in such coarse data resolution. Of the entire series of images on the temporal evolution of geomagnetic variations stored in Supplement (Fig. S2), we will focus here on the most interesting sequence, which lasted from midnight to late morning on 15 May 1921, with special regard to the period from 03:00 UT to 07:00 UT shown in Fig. 5.

Immediately after midnight on 15 May 1921, the mid-latitude observatory De Bilt (IAGA code: DBN) observed a pronounced positive variation in magnetic declination. The other nearby observatories (including Stará Ďala, IAGA code: OGY, shown in red) registered positive variations in declination too, though they were far weaker in the period between 00:00 UT and 01:00 UT. In the course of the following four hours, the whole of this group of observatories already recorded large variations in declination. We may interpret this behaviour of the geomagnetic field as due to the closeness of the outer edge of the auroral

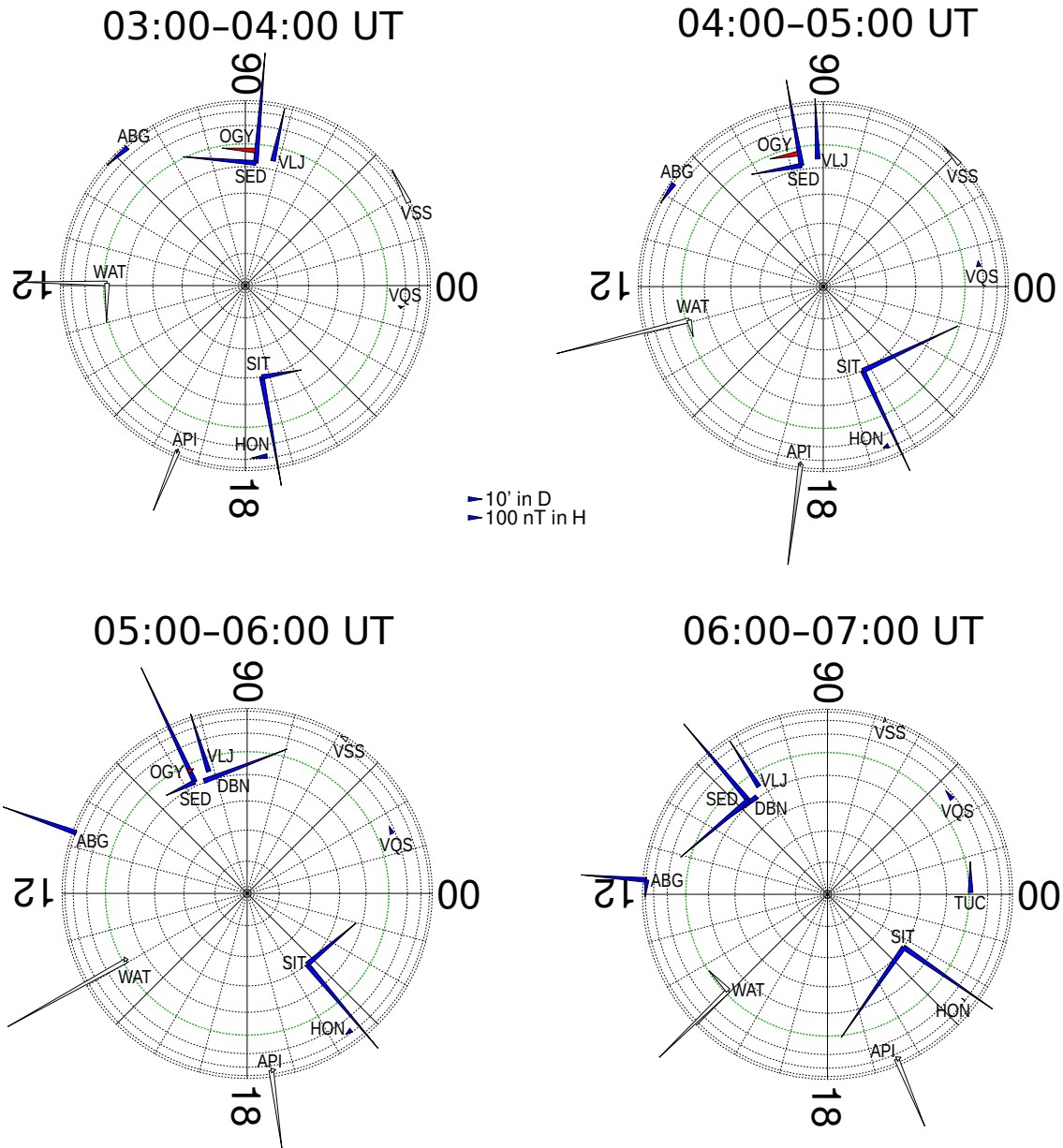

**Figure 5.** Evolution of variations in the horizontal intensity and magnetic declination on 15 May 1921 between 03:00 UT and 07:00 UT. A polar coordinate system is used here: the radial coordinate is the geomagnetic (QDP) latitude, the distance between the individual parallels in the graphs is set to 10°, and the angular coordinate is the magnetic local time. The parallel drawn in green represents the QDP latitude at which the most equatorward observed overhead aurora was reported during the May 1921 storm.

oval. In the same region, the observatory Seddin observed a decrease in vertical intensity. The observatories were apparently
close to the section of the equatorward boundary of the oval associated with the westward electrojet. Judging from the vertical
intensity variations, the outer edge of the oval reached closest to Seddin between 04:00 UT and 05:00 UT (not shown here;
provided as a part of Fig. S2 in the supplement). The horizontal intensity was dominated by the influence of the ring current
throughout the day.

Figure 1 gives us a good reason to suppose that in period 03:00 UT – 07:00 UT, when the vertical intensity was out of
range of the magnetometer, this geomagnetic element exhibited a violent decrease (i.e. negative variation) also in Stará Ďala.
Based on the vertical intensity profile just before leaving the measuring range as well as comparing the profile with the one in
Seddin, we can assume that also Stará Ďala's vertical intensity drop might reach a few hundred nanoteslas, at least sometime
in the period between 04:00 UT – 06:00 UT. Taking the local time into account, we could therefore claim that Stará Ďala was
probably located near the outer edge of the westward electrojet. In addition to the change in the vertical intensity due to the
proximity of the west-flowing currents in the electrojet, the magnetic fields generated by the induced currents in the conductive
ground play an essential role in the variations of the vertical intensity. They are thus strongly dependent on the subsurface
conductivity distribution. Without an appropriate deeper analysis of the subsurface conductivity, which would be beyond the
scope of this study, more detailed interpretations of vertical intensity variations cannot be correctly performed.

On 15 May, during the period from 05:00 UT to 06:00 UT, an interesting situation occurred within the sector 07:00 MLT
– 08:00 MLT (see Fig. 5, bottom left). The deviation of the magnetic declination was then very small at Stará Ďala, almost
negligible; nonetheless, a little more northward located Seddin (in the graphs marked as SED) and De Bilt (DBN) observed
large variations in declination – however, in the opposite direction in relation to each other. It seems to be a local variation that
we can only explain as a consequence of some local electric current. At De Bilt, after a two-hour data gap (probably caused
by an extreme value outside the measurement range of the instrument), the hourly average of declination between 05:00 UT
and 06:00 UT was -12.31°, and the next hourly average was -10.61°, the difference in values being up to 1.7°. Based on the
studied data, we cannot specify the characteristics of this current. Nevertheless, we believe that it was such a sharp and spatially
limited variation, the likes of which typically do not occur in mid-latitudes, and it resembled variations typically occurring in
the vicinity of the auroral oval.

Significant positive declination deviations on the morning side of the Earth persisted until 11:00 UT on 15 May. During the
following hours, the geomagnetic field gradually quietened.

From the arguments above, it seems substantiated to assume that during the whole period 00:00 UT – 11:00 UT on 15 May
1921, at least the group of mid-latitude observatories Val Joyeux (VLJ), De Bilt (DBN), Eskdalemuir (ESK), Seddin (SED),
Stará Ďala (OGY), as well as Prague lay close to the outer edge of the auroral oval. We believe that De Bilt even spent a
while under the auroral oval. In this context, it is worth mentioning that Cid et al. (2015) interpreted the extreme geomagnetic
variations of 29 October 2003 and 2 September 1859. The authors concluded that those extreme variations, which both took
place in the late-morning sector, could be caused by FACs. Similar FACs, which gave rise to additional local currents in the
auroral oval, including the already mentioned nort-south Hall current, seem responsible for the violent variations observed in
the morning sector by mid-latitude observatories on 15 May 1921.

## 4 Discussion and conclusion

The main aim of this study was to reconstruct the observations of the magnificent magnetic storm recorded at the geomagnetic observatory in Stará Ďala on 13–15 May 1921. We achieved this goal through several steps. The very first step was to identify geomagnetic features captured on the records. The original records were not labelled, and the axes were not described at all. By comparing the magnetograms of 13–15 May 1921 with the older records made at the Stará Ďala observatory and its predecessor Ógyalla, as well as with the records from other observatories worldwide, we found that the recorded geomagnetic elements were the magnetic declination and the vertical component of the geomagnetic field (i.e. vertical intensity). Then, we compared the morphological properties of global geomagnetic phenomena in the Stará Ďala record between 13 and 15 May 1921 with the same phenomena observed by other observatories worldwide to assign time marks correctly to the timeline.

In the following, the scale factors were established for the magnetic declination and vertical intensity, with which these elements were plotted on photographic paper. For the declination, we obtained a value of $1.18'/\text{mm} \pm 0.01'/\text{mm}$. This value is comparable with $1.3'/\text{mm}$ published by Valach (2016), who studied Stará Ďala's magnetograms for the storm of 8 March 1918. In the mentioned work, that value was based on data from a single, albeit relatively well-described, magnetogram. The value determined in our study differs from the published value by only $\sim 0.1'/\text{mm}$; nevertheless, we consider our value, determined from a larger number of magnetograms, to be more precisely determined. For the vertical intensity, we obtained a scale of $1.63 \text{ nT/mm} \pm 0.04 \text{ nT/mm}$; we did not compare this scale with another published value, because according to our knowledge, no work has been published about this element for the given period for Stará Ďala.

In the reconstructed magnetograms, there were some rapid changes and relatively large variations in declination (top panel in Fig. 1). For example, on 15 May at around 05:00 UT, the declination change was as much as $51.5' \pm 4.3'$ in less than one hour; this change then continued beyond the measurement range of the instrument. Converted to geomagnetic field units, this change was equivalent to $312.1 \text{ nT} \pm 26.0 \text{ nT}$ in the eastern component of the geomagnetic field (before going out of range). To get an idea of the extreme variation of declination during this storm, let us refer to the quasi-logarithmic scale used in Hurbanovo since 1951 to determine the geomagnetic activity index K (Ochabová, 1955). The boundary between K indices 8 and 9 (i.e. K9 limit) was 350 nT in one of the horizontal components (taking the component for which the deviation was greater). Our value of 312.1 nT is purely a variation in declination (before it goes out of range); the horizontal intensity variation might be even larger.

The data provided by the incomplete vertical intensity record (see Fig. 1, bottom panel) appear to be interesting, too. The decrease in intensity between 22:20 UT on 14 May and 01:22 UT on 15 May was $62 \text{ nT} \pm 2 \text{ nT}$. Subsequently, after a short increase in the value, the intensity dropped again to such an extent that the trace of the beam on the photographic paper got lost, and the instrument went out of the measuring range. The trace returned to the photographic paper (or became visible again) at 07:09 UT (15 May) and increased rapidly in intensity until it reached its maximum at 09:56 UT (15 May); the increase was up to $85 \text{ nT} \pm 2 \text{ nT}$. It was a large variation indeed.

We published the processed magnetic declination and horizontal intensity records for this event in the form of five-minute data in Supplement of this article. Comparison with data from the nearest observatories, namely Prague-Clementinum and Seddin, confirmed that the time series of the reconstructed data show a reasonable course.

We have also shown that the processed data from Stará Ďala can contribute to investigating the global manifestations of the extreme magnetic storm of May 1921. Unfortunately, the horizontal intensity, which is valuable information in the research of geomagnetic activity, was not recorded in Stara Ďala. However, data on the declination and vertical intensity can also help gain valuable insights into the electrical currents of the auroral oval and the associated FACs. Actually, at the time of the peak geomagnetic activity in this event, the auroral oval was expanded far equatorward from its usual location. Stará Ďala, as well as other mid-latitude observatories, thus got close to this complex system of currents; in the morning hours on 15 May 1921 (between 05:00 UT and 06:00 UT), the De Bilt observatory possibly even found itself briefly under the auroral oval.

Such a far equatorward position of the auroral oval and FACs can be compared with the results of the study of other extreme storms. Fujii et al. (1992) found that, during the storm on 13–14 March 1989, the longitudinal width of the FAC region increased, and it did so particularly in the morning sector between 07:00 MLT and 10:00 MLT. Such a phenomenon might be related to the variations we have shown for the same morning sector in Fig. 5. When comparing the storm from 29 October 2003 with the superstorm on 2 September 1859, Cid et al. (2015) pointed out probable FACs peaks within the same prenoon sector (∼9 MLT).

By studying FACs observed with the satellite CHAMP during the intense 2003 geomagnetic storm events, Wang et al. (2006) determined the most equatorward magnetic latitudes of FACs. In the dayside (close to noon meridian) in the summer (southern) hemisphere, the minimum magnetic latitudes for October 2003 events were comparable with the QDP latitude of the De Bilt observatory (not taking the sign into account because of the opposite hemispheres). As we indicated in our study, the equatorward boundary of the auroral oval may have reached there for a while during the storm in May 1921. Our finding is thus consistent with the conclusion of Wang et al. (2006) that in extreme geomagnetic storms, the minimum magnetic latitude of FACs associated with the auroral oval is saturated at ∼50° (Wang et al. states 52°).

*Data availability.* The resulting digital data of the magnetic declination and the vertical intensity for Stará Ďala on 13–15 May 1921 are available in Supplement.

*Author contributions.* EK determined scale coefficients for the old magnetograms of Stará Ďala, visualized global geomagnetic variations with a series of polar graphs and participated in their interpretation, and took part in writing the text. FV devised the idea of the study, digitized five-minute and momentary data from the old magnetograms of Stará Ďala, and participated in the interpretation of geomagnetic variations and in writing the text.

*Competing interests.* The authors declare that they have no conflict of interest.

*Acknowledgements.* The research presented in this paper was funded by the Scientific Grant of the Ministry of Education of Slovak Republic and the Slovak Academy of Sciences, grant VEGA no. 2/0085/21. Thanks also go to climatologist RNDr. Dalibor Výberči, who brought to our attention the paper about the aurora borealis in the Czech Republic. We express our gratitude to all three referees for their valuable comments aimed at improving our manuscript. One of the anonymous referees advised us to use an amount of north-south Hall current to interpret the declination variations, for which we are very grateful. Special thanks go to Dr. Hans-Joachim Linthe for providing us with hourly means and magnetograms of the Seddin observatory for the studied event. We are grateful to the staff of the Niemegk Observatory, which preserves data from the historical observatories of Potsdam and Seddin, the predecessors of the Niemegk Observatory. We acknowledge all the observatories who made the data used in this study available via the world data centres. Also, we thank the WDC for Geomagnetism, Kyoto, for collecting the geomagnetic data.

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
