# Peer review of "The record of the magnetic storm on 15 May 1921 in Stará Ďala (present day Hurbanovo) and its compliance with the global picture of this extreme event"

_Annales Geophysicae, 2023_

## Author Response (AR1)

**Response to Referee #1**

*Authors: We would like to thank Referee #1 very much for his or her time and work devoted to carefully reviewing our manuscript and for the helpful comments.*

The paper introduces newly recovered historical data contributing to the investigation of the 13-15 May 1921 extreme geomagnetic storm. Since both extreme geomagnetic events and their surviving recordings before the 1930s are rare, any new data are important. The unscaled recordings were carefully calibrated. The authors presented the Stara Dala data in the context of other observations of the same storms, as well as closely located aurora observations. The section detailing data processing is too lengthy.
*Response:*
*The data calibration for the preserved magnetogram forms an important part of our manuscript because in this part of our study we tried to piece together the sketchy information in order to obtain the most reliable resulting data available. We have tried to make the manuscript more attractive by adding a piece of information about the content of the relevant parts of the text. Namely, in Lines 89–91 (line numbers in the manuscript with the changes marked) we clarified that the Section 2.2 describes in detail the procedure of the calibration, and the resulting calibrated data are presented later, in Section 2.3. By such improved navigation in the text, we will hopefully allow the reader who is less interested in the calibration procedure to avoid the uninteresting parts.*

Global magnetic data used for this study re taken from WDC. However, it is not clear if this is a complete set of the available data. Figures 4 and 5 includes 10-12 data points, while Hapgood listed 21 observotories (excluding Stara Dala) with available data.
*Response:*
*The data utilised here were all the hourly means of the geomagnetic field available for the period in question via the webpage of the World Data Center for Geomagnetism, Kyoto. The hourly means of Seddin were obtained from the Niemegk observatory, the successor of the Seddin observatory. (Lines 296-299.)*

The authors seem to consider only Czech aurora observations. Hapgood mentions other sources (see also their references). Relatively little information is given on how other researchers interpreted the available data related to this event. Missing the summary of others findings it is difficult to judge what are the new findings of this paper.
*Response:*
*In our work, we wanted to point out to the international scientific community a little-known piece of information about the aurora borealis observed during the storm of May 1921 from the territory of former Czechoslovakia. In the new version of the manuscript, we have also added that the aurora observations from former Czechoslovakia extend the information about observed auroras to a part of Europe not covered in Hapgood's review article. Also, we have mentioned how the observations from Czechoslovakia fit into the information published so far (Lines 251 – 256).*

Minor comments:

18: curtailed > limited
*Response: Corrected. (Line 18 in the manuscript with the changes marked)*

29: size > magnitude
*Response: Corrected. (Line 34)*

38: extent > the extreme equatorward extension
*Response: Corrected. (Line 43)*

41: an extreme solar storm > the causative solar storm
*Response: Corrected. (Line 46)*

42: Mountwilson > Mount Wilson
*Response: Corrected. (Line 47)*

47: will be > could be
*Response: Corrected. (Line 52)*

52: in low magnetic latitudes > at low magnetic latitudes
*Response: Corrected. (Line 57)*

63: the descending phases > a declining phase
*Response: Corrected. (Line 68)*

Table 1: Throughout the paper CGM latitudes are used, geomagnetic coordinates could be removed from the table. They just confuse the reader. See also comment to line 277.
*Response: CGM coordinates have been changed to QDP coordinates. (Lines 244-245, 248, 305-306-307, 329 Table 1, Table S1, caption of Figure 4)*

Table 2 and throughout the text: The unit in this form is not correct as it includes a decimal dot. Use „arc minute/mm" or „'/mm" instead.
*Response: All the units that had been improperly included a decimal dot have been corrected in the new version of the manuscript. (Lines 152, 156, 158, 170-172, 410-411 and 413, and Table 2)*

162: centre > median (?) Please specify clearly!
*Response: The word "centre" has been changed to "median" (Line 170).*

175: Contemplating > Considering
*Response: Corrected. (Line 183)*

182: thoroughly selected: based on what selection criteria?
*Response: We selected records in which the light trace on the magnetogram (photo paper) was clear and thin, to allow obtaining values with as little uncertainty as possible (Lines 191-192).*

Table 5 and text: Use nT/mm as scale unit!
*Response: All the units that had been improperly included a decimal dot have been corrected in the new version of the manuscript. (Lines 202, 208 and 415, and Table 5)*

201: we read the five-minute data (i.e., five-minute means): this needs to be clarified. Means are typically calculated and not just read.
*Response: We added an explanation, "For this purpose, we imitated the method commonly used in determination the hourly averages by means of a glass scale. The mean value was determined using an imaginary horizontal straight line, which was put on the magnetogram so that we were making equal the areas between the trace being scaled and the horizontal line (McComb, 1952, pp. 177–178)" [see Lines 211-214]. In the Reference, we added one new item, namely (McComb, 1952) [Line 509].*

202: not at our disposal > missing OR Where the magnetograms were incomplete
*Response: We have used "Where the magnetograms were incomplete…" Thank you for the suggested wording. (Line 215)*

Table 6 caption in … days > on … days
*Response: Corrected.*

Figure 1: Use wider line, connect sporadic red points with a thin or dotted line to guide the reader eye (line in Fig 2). Now it is difficult the follow the development of the storm in the presented plots. Consider if you could make this figure smaller.
*Response: We have made the figure smaller, widened the line and connected the sporadic red points.*

Figure 2: This figure could be made smaller. Note in the caption that the lines connecting the observations is added only to guide the reader's eyes.
*Response: The note has been added to the caption of Fig. 2. The figure has been made smaller. The questionable values composing the second maximum are marked with question marks.*

226: limpid > clear
*Response: In the new manuscript, we have used the word "transparent". (We like the word "clear" as well, but it had already been used in the same sentence.) [Line 241]*

226: sank > set
*Response: Corrected. (Line 241)*

227: the light was whitish even white: This is not very clear. maybe 'whitish or even white'
*Response: We have added "or". (Line 242)*

243: positive: does it have any relevance? The baseline is instrumental and does not have any physical meaning.
*Response:We have erased "positive" and only written "eastern". (Line 267)*

Fig 3: Thicker lines, smaller figure with properly adjusted caption size.
*Response: A completely new Figure 3 has been included in the manuscript, in which we show original analogue magnetograms recorded by the Seddin observatory.*

273: reference point > origin or pole
*Response: Corrected   we have used "origin". (Line 305)*

277: Use QDP coordinates everywhere rather than a mixture of coordnates (it is unlikely that you need to redraw any plots because of this technical correction, since as you mentioned QDP and CGM are indistinguishable at high latitudes. Modify Table 1 accordingly.
*Response:*
*Table 1 has been modified (quasi-dipole coordinates have been listed).*
*The QDP coordinates have been used everywhere in the new version of the manuscript.*

283: the orientation of the arrows is adjusted as the respective variations would appear in the northern hemisphere if they were the result of currents in auroral ovals or FACs belonging to the northern hemisphere: This is not very clear to me.
*Response:  An explanation is provided in Lines 315-316 and 319-322.*

288: parallels > circles of latitude
*Response: Corrected. (Line 325)*

290: increase: of what? Help the reader!
*Response: Increase in the vertical intensity. Added to the text. (Line 327)*

294: outer: ok, but equatorward is more specific.
*Response: Changed to "equatorward". (Line 331)*

318: locality > region
*Response: Corrected. (Line 362)*

320: close to the equatorward boundary of the oval, in this part of which the westward electrojet flowed > close to the section of the equatorward boundary of the oval associated with the westward electrojet [or sg similar]
*Response: We have used the offered wording. Thank you for it. (Lines 364-365)*

329: A possibly interesting … : Rephrase this sentence, make it shorter and simpler.
*Response: Rephrased (Lines 380-381).*

332: each to other > in relation to each other
*Response: Corrected. (Line 384)*

333 With a great deal of… : Rephrase this sentence [themselves, northernmost]. This is a statement, the reasoning is missing.
*Response: We have rephrased this sentence. In the current manuscript, the new piece of text is in Lines 386–391.*

Figure 5. There is no need to repeat almost the whole caption of Fig 4. You could say simply: Same as Fig. 4 but for the vertical component.
*Response: This figure has been removed from the manuscript.*

340: good to mention > worth mentioning
*Response: Corrected. (Line 397)*

346: To achieve this goal, we dealt with several partial tasks > We achieved this goal through several steps.
*Response: Thank you for the improved wording. We have used it in the new text (Lines 403-404).*

355: can be compared > is comparable
*Response: Corrected. (Line 413)*

Units!
*Response: Corrected.*

365 and 377: You already mentioned in line 15 the changes of the name of the location throughout its history. It is ok to remind the reader once but it is absolutely unnecessary to do it twice
*Response: We have removed this unnecessary information from here (Lines 423-424 and 434-435).*

378: 56.5 nT: Not clear why this arbitrarily chosen temporal variation is relevant for the storm studied.
*Response: This part of the text has been removed  (Line 436).*

**Response to Referee #2**

Review comments on the manuscript 10.5194/angeo-2023-12
" The record of the magnetic storm on 15 May 1921 in Stará Dala (present day Hurbanovo) and its compliance with the global picture of this extreme event"
E. Koci and F. Valach

*Authors: We would like to thank Referee #2 for his or her time and work devoted to reviewing our manuscript and for the helpful comments.*

The authors have tried to reconstruct the historical magnetic field recordings of the former Stará Dala observatory for the period of the major magnetic storm in the middle of May 1921. This can be considered a valuable contribution to better characterize space weather conditions during extreme magnetic storms. By applying different approaches for recalibrating the recordings and by comparison with neighboring observatories they obtain reasonably convincing data series for the time period of interest.

Besides these generally positive results the study contains also weaknesses, in particular when it comes to the storm-related magnetic variations. More details of the expected improvements are listed below. Overall, the work is regarded worth being made public, but substantial revisions are expected before it should appear in Annales Geophysicae.

General comments

1) The interpretation of the storm features based on the observed magnetic variations is not convincing. Generally, it is a pity that the northward, H component is missing at Stará Dala. This is most important for the characterization of magnetic activity. Deflections of the vertical, Z component are strongly influenced by the subsurface conductivity distribution. This fact should be clearly stated in the paper. For example, the observatories Wingst and Niemegk exhibit commonly opposite deflections during times of magnetic activity. This is caused by the effect of the so-called northern German anomaly. After having said that, it is worth to continue with the available data from Stará Dala.

*Response: In Lines 374-378 (line numbers in the manuscript with changes marked) we added the following explanation: "In addition to the change in the vertical intensity due to the proximity of the west-flowing currents in the electrojet, the magnetic fields generated by the induced currents in the conductive ground play an essential role in the variations of the vertical intensity. They are thus strongly dependent on the subsurface conductivity distribution. Without an appropriate deeper analysis of the subsurface conductivity, which would be beyond the scope of this study, more detailed interpretations of vertical intensity variations cannot be correctly performed."*

In the paper the shown variations recorded at Niemegk give the most complete picture of the storm evolution. However, they are taken about 5° north of Stará Dala. This can make significant differences during a magnetic storm. It would have been very instructive to add complete field recordings from similar latitudes. In my view Munich-Bogenhausen could provide valuable recordings for comparison, complementing very well the Niemegk data. The H variations from Munich could help to better quantify the southward extend of the electrojet.

*Response: We have added the following explanation in the new manuscript: "Unfortunately, the relevant hourly means in Munich are not available as recording the temporal variations of the*

*geomagnetic field were discontinued during the years when the storm we are interested in occurred (Soffel, 2015)." (Lines 24-25) Also, a new reference has been added: H. C. Soffel, History of the Munich–Maisach–Fürstenfeldbruck Geomagnetic Observatory, Hist Geo Space Sci, 6, 65–86, 2015.*

2) Another topic of concern is the frequent quoting of field-aligned currents (FAC) in connection with observed eastward, D component variations. It is known since Fukushima's famous publication that field-aligned currents cause virtually not magnetic signature on ground. For that reason, all the parts where FACs are mentions should be revised in this respect. If FACs are quoted, the related Hall currents, that give rise to ground deflections, have to be introduced and made consistent with the observations. In some parts the major storm of October 2003 is taken as reference. For that, detailed observations from ground and satellite are available. For example, Wang et al. (Annales Geophysicae, 24, 311–324, 2006) describes well the relation of FAC to the intensity of solar wind input and ring current activity. This is different for day and nightside. Considering their results may help to support the offered interpretation.

*Response: We have removed some interpretations related to this comment from Section 3.2.2 and from Discussion. Instead, we focus on the declination variations that were observed in the mid-latitudes in the morning sector between 03:00 UT and 07:00 UT. However, we do not quite agree with the referee regarding the application of Fukushima's Theorem to middle geomagnetic latitudes. Because the geometry of the geomagnetic field in mid-latitudes does not strictly meet the assumptions of Fukushima's Theorem (field lines in mid-latitudes cannot be considered perpendicular to the earth's surface; moreover, in disturbed geomagnetic conditions, the assumption of uniform conductivity might not be well fulfilled), even ground-based geomagnetic observations can capture at least some manifestations of FACs. In our opinion, this may also be the case for the mentioned mid-latitude declination variations in the morning sector between 03:00 UT and 07:00 UT. We added a paragraph that explains it in Lines 345–350. Side note (not included in the manuscript): Fukushima in another of his works "Field aligned currents in the magnetosphere" (in Geofísica Internacional, 1991, 30/4, pp. 241-248) dealt with field-aligned currents at middle and low latitudes. He wrote, for instance, that "The seasonal variation of the Sq field (in particular for magnetic declination or Y-component) will be attributable to some field-aligned currents […]". However, these are not the FACs related to the auroral oval, of course.*

Detailed comments
Line 117: The sentence "We also know that the then device worked in …" is not clear.
*Response: The recording in the variation device was made on photographic paper, which was moved at a speed of 1 cm per hour using a clockwork machine. The explanation has been added in Lines 125-126 (line numbers in the manuscript with the changes marked).*

Sect. 2.2.3: The variations of the vertical component depend strongly on the subsurface conductivity; opposite deflections are observed between Wingst and Niemegk. The dependence on conductivity has to be mentioned.
*Response: Done in Lines 374–378.*

Fig. 1 It would have been instructive to mark the times of SCs here, possibly also in Fig. 3.
*Response: We have marked the times of SCs in Figs. 1 and 3.*

Fig. 2 The second peak in declination from Clementinum is rather questionable, while the first corresponds reasonably well with the related variations in Niemegk. However, around 18 UT the storm activity has died out. It is thus practically impossible that such a large D deflection could have happened. Here again the Munich data could be decisive. When presenting historical data, it is important to check critically their reliability. This critical assessment of the Clementinum data has to be spelled out clearer, e.g. in the paragraph following line 240.

*Response: We also admit the two unexpectedly extreme data forming the second maximum might be erroneous; the cause of the error, however, has remained unrevealed for us. We added such an idea in Lines 270-271. In Fig. 2, we also marked graphically (using question marks) the two data in question.*

Line 245ff: "Possibly it might be a manifestation of a field aligned current", All these statements concerning FACs in relation to declination variations are misleading. They should be removed here and elsewhere.

*Response: See our answer to the general comment no. 2.*

Lines 320ff: The relations between H und Z variation at Niemegk and possibly Munich, could be used to estimate the latitudinal position of the westward electrojet. Actually, Z goes through zero under the electrojet. A simple model for quantifying the H to Z relation is to assume a line current in the ionosphere.

*Response: Yes, we agree. However, this should also involve analysing subsurface conductivity, dealing with spectral analysis of the geomagnetic variations etc, which would be beyond the scope of our study.*

Fig. 5: I am not sure what to learn from this figure. The sign of Z deflection varies from place to place, and not conclusions are drawn in this work from it. It well could be dropped or put into the Supplements.

*Response: We have discarded Figure 5, which had shown the variations in the vertical intensity, from the new manuscript. The series of the images are only kept in the supplement.*

Lines 341ff: In comparison with the 29 Oct. 2003, storm the Wang et al. paper should be used to help quantifying the expansion of the auroral oval. Useful information can be obtained in this regard from a reconstruction of the ring current intensity from low-latitude stations.

*Response: We have compared the results of Wang et al. (2006) for the day side close to the noon meridian with the QDP latitude at which we expected the most equatorward extent of the auroral oval in the morning sector on the morning on 15 May. Our results seem to agree with what they found for the October 2003 events (Lines 456-462).*

Lines 388ff: The conclusions listed here are presently pure speculation. After revision of the manuscript, they have to be improved.

*Response: The conclusions have been revised (Lines 450-462).*

**Response to Referee #3**

Referee on Eduard Koci and Fridrich Valach: The record of the magnetic storm on 15 May 1921 in Stará Dala (present day Hurbanovo) and its compliance with the global picture of this extreme event

The authors contribute with this article a data set of the strong magnetic storm of 13-15 May 1921, which was up to now not yet considered. The publication extends with this important data set the possibilities of researching an interesting geomagnetic event. The authors found a suitable method to recalibrate the data. For my opinion the authors succeeded in this comprehensive challenge. I recommend this article for the publication in Anales Geophysicae. Only minor changes are necessary.
*Response: We would like to thank Dr. Linthe very much for his time and work devoted to our manuscript and for the comments. We are especially grateful for the offered magnetograms from Potsdam and Seddin as well as for the hourly means of Seddin.*

I have the following detailed comments:

Caption and last line of table 1, line 238 and further lines: At the time of the magnetic storm the observatories Potsdam and Seddin existed, but not Niemegk. Niemegk was started to be established only in 1929 and opened in 1932. Therefore, Niemegk should be skipped or only mentioned as the successor station of Potsdam and Seddin.
*Response: In the new manuscript, the magnetograms and hourly means of Seddin have been used. (Lines 27-30, 79, 227, 258, 264, 268, 274, 372, 383, 395-396, 442 and Table 1)*

Line 107: Supplement ( S1) – There is no supplement and no Fig. S1
Line 281: Supplement ( S2… – There is no supplement and no Fig. S2
Line 205 and further lines: supplement – There is no supplement
*Response: The supplement, which is containing Figs. S1 and S2, is now available on the following address: <https://angeo.copernicus.org/preprints/angeo-2023- 12/angeo-2023-12-supplement.zip>*

Line 207 and further lines: The word "registration" should be changed into "recording"
*Response: We have changed the word "registration" to "recording" or "record" throughout the manuscript. (Lines 124, 210, 221, 266)*

Caption of Fig. 3: A better word for "one-hour means" is "hourly means" or "hourly averages".
*Response: Throughout the manuscript, we have changed words "one-hour means" to better words, as recommended. (Lines 276, 286, 297, 300, 302, 351)*

The data of Potsdam and Seddin distinguish by several nT, especially during geomagnetic storms. Therefore, it is not common to present a plot of "Potsdam/Seddin". You need to plot the data of Potsdam or [Seddin.] Or, if you averaged the data of both observatories to mention about this.
[Fig.] 3: A plot of hourly mean values is not very valuable due to the ramps every hour. I could offer you analogue copies of the original magnetograms of both observatories Potsdam and Seddin, which are available in the Niemegk data archive. It is up to you to decide, what is better to be used. You may get an image of such copies in:
Linthe, H.-J.: History of the Potsdam, Seddin and Niemegk Geomagnetic Observatories – First Part: Potsdam, History of Geo- and Space Sciences, submitted 2022
*Response: We have changed the plot of "Potsdam/Seddin" hourly means to the analogue copy of the original magnetogram recorded by the observatory Seddin.* We are grateful to Dr. Linthe for providing the analogue copy for our manuscript.

---

## Referee Report (RR1)

The revision improved the manuscript considerably. I agree completely in the content of the manuscript. I recommend to publish the paper as it is.

Hans-Joachim Linthe

Volunteer

Helmholtz Centre Potsdam – GFZ

Niemegk Adolf Schmidt Geomagnetic Observatory

---

## Author Response (AR2)

**The authors' response to Referee #1**

Comments on Reply on RC2', of Fridrich Valach, 30 May 2023

I am generally satisfied with the authors' responses to my comments.

There is one exception. I cannot agree with their statement that an appreciable part of the eastward By component deflections are caused by field-aligned currents. It is true that the Fukushima theorem is strictly valid only at polar regions. It has to be noted, however, at European latitudes the field inclination is about 70°, which is much closer to the polar than to the equatorial conditions. If the authors insist on their claimed FAC to By relation, they have to demonstrate how much FAC density and which geometrical configuration is required for generating such large By deflections. Also, the influence of conductivity gradients is often overestimated in the context of FAC ground effects. Such hand waving arguments are not sufficient.
Much more convincing and easier to explain is a certain amount of Hall current in north-south direction for causing the By deflections. Such north-south Hall currents appear between pairs of FACs separated in east-west direction. This FAC issue definitely has to be fixed before publication.

*Response:*

*After we have rethought the issue of FACs vs declination variation, we agree with the referee.*

*Indeed, from a simple model in which we imagine the FAC as part of a very long straight wire, which must be above the dynamo layer, and assuming an inclination of 70˚, it follows that:*
*A. The magnetic field due to the FAC can be recorded only by ground-based observatories that are not less than 300 km away or even farther away.*
*B. For the 2003 event, Wang et al. (2006) found FACs with current densities of no more than 10 $\mu A/m^2$. Assuming a cross-section of the wire of about 10,000 square kilometres (just our guess), it would give a current of $10^5$ A. Such a current could cause a variation of 70 nT at a distance of 300 km. For the horizontal intensity of 20,000 nT, those 70 nT mean a declination change of 12'. This value is approximately one order of magnitude less than the observed change in declination (hourly means).*
*Based mainly on item A, we realize that our interpretation needs to be corrected. We are grateful to the reviewer for his/her advice on Hall current in the north-south direction between pairs of FACs separated in the east-west direction.*

*The changes we have made in the manuscript are as follows:*
*(We refer to the line numbers in the manuscript with indicated changes.)*

*Lines 260-261 have been removed.*

*Line 306: "or FACs" have been erased.*

*Lines 324-336 have been replaced with a new paragraph (Lines 337-343).*

*Line 354: A part of the text in this line has been erased.*

*Lines 386-387: The text in these lines has been changed.*

*Lines 467-468: One item in References was removed.*

We thank the editor Dr Christos Katsavrias and the anonymous referee for their time devoted to our manuscript.

---

## Author Response (AR3)

**The authors' response to the Referee**

Comments on Re-reply to RC2', of Fridrich Valach, 19 July 2023

I am much satisfied with the authors' responses to my main concerns.
However, in this new version there appear a number of weak points and flaws that need to be corrected before publication.

Line 205: The diurnal Bz variations at your station in summer are probably caused by the Sq currents. However, the amplitudes of those variations depend strongly on the solar flux. Are the solar activity conditions the same in 1910 and 1921?.
Owolabi et al. (2022) https://doi.org/10.1029/2021JA029903, showed in their Figs. 2 and 3 the strong dependence of Sq intensity on F10.7. For your period no F10.7 is available, but the sun spot number can also do a good job in this respect. This kind of dependence should be taken into account for the Bz scale factor or at least be discussed.

*Response:*
*We investigated the relationship between daily sunspot numbers and the increase in the vertical intensity in millimetres. In agreement with Table 6, we performed this analysis for two separate periods: (1) for the data of 1909 and 1910 and (2) for the data of 1921. The data and the calculated correlation coefficients are listed below.*

| *Day* | *Sunspot number* | *Increase in Z (mm)* |
|---|---|---|
| *16/07/1909* | *43* | *3.05* |
| *25/07/1909* | *153* | *2.90* |
| *23/05/1910* | *38* | *2.90* |
| *04/06/1910* | *45* | *3.45* |
| *06/06/1910* | *12* | *2.20* |
| *07/06/1910* | *28* | *2.70* |
| *14/06/1910* | *10* | *2.85* |
| *30/06/1910* | *30* | *3.20* |
| *01/07/1910* | *17* | *2.80* |

*Correlation coefficient:      0.216   (i.e., very weak or no relationship)*

| *Day* | *Sunspot number* | *Increase in Z (mm)* |
|---|---|---|
| *23/05/1921* | *18* | *5.70* |
| *25/05/1921* | *22* | *5.00* |
| *01/06/1921* | *42* | *5.50* |
| *05/06/1921* | *33* | *5.55* |
| *18/07/1921* | *22* | *5.85* |

*Correlation coefficient:      -0,062   (i.e., no relationship)*

*The two periods fell into the declining phase of the two consequent solar activity cycles. The sunspot number did not differ substantially between the selected days in 1909-1910 and 1921. Based on this short analysis, we assume that our determination of the scale factor was not influenced substantially by different conditions in solar activity in those two periods.*

*This analysis is summarized in Lines 210–219. References were added in Lines 499 and 504–506. (Line numbering is according to the manuscript with indicated changes.)*

Fig. 1: I am confused by your lines marked, SCs. I the Introduction you mention three SCs: 13 May, 13:06 and 19:24 UT, 14 May, 14:05 UT. The first line may well mark the 13:06 UT event, but for the second, on 14 May, I find no correspondence in your list.

Your dealing with SCs, in general, is quite confusing for me. When comparing the listed times with the recordings at Sedin I can confirm the times listed for 13 May but not for the following day. Rather, a clear SC signature appeared at 16:0x UT, not at 14:05 UT, and there is definitely no SC at 22:10 UT, where the mark appears. The time of an SC should not be confused with the start of magnetic activity, which typically comes a few hours later. Please check again and correct the SC occurrences. I would love to see correct lines for all three SCs in both figures.

*Response:*
*In the previous version of the manuscript, we had awkwardly combined information from three sources (namely; Lundstedt et al., 2015; Hapgood, 2019; and Mayaud, 1973). In addition, we made a typo that made everything even worse. We apologize for the confusion.*
*In the new version of the manuscript, we have corrected all information about the times of occurrence of the SCs according to the primary source, which is Mayaud (1973). The SCs mentioned in the manuscript are all those identified by Mayaud. Figs. 1, 3, and 4 now display three SCs. The corrections regarding SCs are in Lines 58–61, 65–67, 229–231, 287–289. The captions of Figs. 1, 3 and 4 have been updated as well.*

Line 267: In the present version no hourly means of the Sedin observatory appear. It would be highly desirable to add that former Fig. 3 to the final version as Fig. 4 because it well outlines the gross features of that storm. Without such an overview plot the global variations shown in the figure below are difficult to understand.

*Response: Done. (Fig. 4 as well as Lines 270–273, 276, 283–284, and 287)*

After removing these weaknesses, the manuscript is regarded as suitable for publication in Ann. Geophys.

**The authors thank the anonymous Referee for his or her help in improving the manuscript.**